# Application of Phosphate Materials as Constructed Wetland Fillers for Efficient Removal of Heavy Metals from Wastewater

**DOI:** 10.3390/ijerph19095344

**Published:** 2022-04-27

**Authors:** Xiaodan Wu, Ni Hong, Qingjing Cen, Jiaxin Lu, Hui Wan, Wei Liu, Hongli Zheng, Roger Ruan, Kirk Cobb, Yuhuan Liu

**Affiliations:** 1State Key Laboratory of Food Science and Technology, Engineering Research Center for Biomass Conversion, Nanchang University, Ministry of Education, Nanchang 330047, China; wuxiaodan@ncu.edu.cn (X.W.); 412345420146@email.ncu.edu.cn (N.H.); 402313319002@email.ncu.edu.cn (Q.C.); wanhui_4331@163.com (H.W.); honglizheng@ncu.edu.cn (H.Z.); 2State Environmental Protection Key Laboratory of Food Chain Pollution Control, Beijing Technology and Business University, Beijing 100048, China; lujiaxin@btbu.edu.cn; 3Agricultural Products Testing Sub-Center, Nanchang Inspection and Testing Center, Nanchang 330005, China; liuwei5055@126.com; 4Center for Biorefining, Department of Bioproducts and Biosystems Engineering, University of Minnesota, St. Paul, MN 55108, USA; ruanx001@umn.edu (R.R.); kcobb@umn.edu (K.C.)

**Keywords:** copper, physic acid sodium salt hydrate, adsorption selectivity, adsorption stability, adsorption kinetic model

## Abstract

Constructed wetlands are an environmentally friendly and economically efficient sewage treatment technology. Heavy metals (HMs) removal is always regarded as one of the most important tasks in constructed wetlands, which have aroused increasing concern in the field of contamination control in recent times. The fillers of constructed wetlands play an important role in HMs removal. However, traditional wetland fillers (e.g., zeolite, sand, and gravel) are known to be imperfect because of their low adsorption capacity. Regarding HMs removal, our work involved the selection of prominent absorbents, the evaluation of adsorption stability for various treatments, and then the possibility of applying this HM removal technology to constructed wetlands. For this purpose, several phosphate materials were tested to remove the heavy metals Cu and Zn. Three good phosphates including hydroxyapatite (HAP), calcium phosphate (CP), and physic acid sodium salt hydrate (PAS) demonstrated fast removal efficiency of HMs (Cu^2+^, Zn^2+^) from aqueous solution. The maximum removal rates of Cu^2+^ and Zn^2+^ by HAP, CP, and PAS reached 81.6% and 95.8%; 66.9% and 70.4%; 98.8% and 1.99%, respectively. In addition, better adsorption stability of these heavy metals was found to occur with a wide variation of desorption time and pH range. The most remarkable efficiency for heavy metal removal among tested phosphates was PAS, followed by HAP and CP. This study can provide a basis for the application of HMs removal in manmade wetland systems.

## 1. Introduction

In recent years, with the increasing attention paid to wastewater treatment, constructed wetlands have been widely developed as successful strategies for reducing contaminant loads, especially heavy metals (HMs) removal, from manufacturing and wastewater effluents [1,2]. The primary mechanisms for high-efficiency phytoremediation removal can be attributed to biosorption [3], biotranslocation [4], bioaccumulation [5], and inactivation by plant roots. However, high concentrations of accumulated HMs in diverse plant tissues probably induce toxic effects on hydrophytes, and even pose a potential threat to their growth [6,7], particularly for hyper-accumulator species [8]. However, most HMs still remain in the wetland systems; the HMs can only be removed by actually harvesting plants into which they have been absorbed. After that, how to ensure the safe utilization of the aquatic plants is also a challenge. Therefore, removing hazardous metals in the early stage is a key factor to be considered in the construction of wetlands. 

Another problem that should not be overlooked is heavy metal pollution in wetlands soil. With excessive human activities, heavy metal pollution aggravates the deterioration of wetland environments [9,10], and has increasingly caused a large burden on pollution treatment and management. Various wetlands have been polluted by HMs, such as urban wetlands [11], riparian freshwater wetlands [12], and reclaimed wetlands [13]. Wetlands play an irreplaceable role in the environment [14] and should be well protected. Multiple effluents, such as industrial discharge, urban wastewater, sewage sludge, animal manure, and landfill leachate, are some of the main sources of HMs contaminants. Therefore, controlling HMs pollution from these wastewaters is also an urgent task for constructed wetlands.

In terms of HMs pollution treatment in the environment, plenty of superior techniques have been developed, where adsorption is well known as a high-efficiency and low-cost method [15]. Phosphates are widely found in nature and are used in the passivation of heavy metals in an inexpensive and effective manner. Among various adsorbents, phosphate-based materials have been shown to be one of the most effective methods for HMs removal from wastewater [16]. Natural phosphate and modified phosphate materials are more attractive than activated carbon or other synthetic materials, which are unavailable and uneconomical in large-scale applications [17]. Different phosphate species exhibit a variety of HM removal roles, owing to their individual properties [18], in particular, porosity and specific surface area, and PO_3_^4−^ in phosphate materials induces stable phosphate precipitation of heavy metals in soil, thus reducing the effective concentration of heavy metals and reducing the bioavailability of heavy metals. It is often used for remediation of Pb, Cd, Cu, Zn, and other elements [19]. Iconaru et al. [20] found that hydroxyapatite nanoparticles have a great capacity to adsorb lead (II) from aqueous solutions. Various phosphate materials, such as bone meal, phosphate rock, and super phosphate, have been used as effective passivators in HMs immobilization in soil remediation [21]. Cao et al. [22] found that phosphate can effectively reduce the biological effectiveness and leaching rate of Pb in contaminated soils by forming stable phosphoric chlorite (Pb_5_(PO_4_)_3_(Cl/F/OH)). Wang et al. [23] repaired the tailings site with phosphate such as apatite, and after 90 days of treatment, the available states of Pb, Cd, and Zn decreased by 22–81%, 15–31%, and 12–75%, respectively. There are few studies on the application of phosphates for constructed wetlands. Hence, more testing is needed to confirm that phosphates can be used as alternative materials for metal removal in constructed wetlands so that phosphates can be both applied to purify wastewater and remedy contaminated soil in wetland systems. Therefore, the development of heavy metal pollution treatment with phosphate materials is worthy of consideration, with a reasonable prospect for success. 

Data of adsorption equilibrium, kinetic rates, and stability are significant to the design the wetland systems and predict operational effects. For this study, several phosphate materials were tested to capture and remove HMs from aqueous solutions, with this future application in mind, to provide theoretical support for wetland adsorption systems. The objectives were as follows: (a) determinate the removal efficiency of HMs onto phosphates and pick out useful materials for HMs treatment; (b) analyze adsorption kinetic models of HMs onto phosphates; (c) explore the adsorption selectivity and stability of HMs; and (d) design a wetland adsorption system using phosphate materials and evaluate its operational effects.

## 2. Materials and Methods

### 2.1. Materials

Trial reagents contained the following substances: copper chloride (AR grade, 98% purity), zinc chloride (AR grade, 98% purity), calcium phosphate (CP), calcium phosphate hydrate (CPH), calcium phosphate monobasic monohydrate (CPM), phytic acid sodium salt hydrate (PAS); these items were available from Shanghai Macklin Biochemical Co., Ltd., Shanghai, China. Hydroxyapatite (HAP) was obtained from Shanghai Hualan chemical technology Co. Ltd., Shanghai, China. Another material, bone meal (BM), was available from Qinghe ceramic raw materials store, Jingdezhen, China. All of these experimental materials and their chemical characteristics are listed in Table 1.

### 2.2. Batch Adsorption Experiments

Initially, removal performances of copper on several phosphate materials were tested. Six materials (HAP, BM, CP, CPM, CPH, PAS) were screened with uniform particle size (0.150 mm) and added on the basis of mole ratio (phosphate: metal = 0.12:1) into 1000 mL copper chloride solution, respectively. All the groups were homogeneously stirred and covered with plastic wrap and then incubated statically at room temperature. A small volume sample of each solution for each treatment was pipetted into a 50 mL plastic centrifuge tube at 0, 6, 12, 24, 48, and 72 h, respectively. Then, the concentrations of Cu in each sample were tested according to the method described by Zhou et al. [24].

The adsorption performances of these materials were assessed by the removal rate of HM, as calculated by Equation (1).
(1)$$R=\left(1-\frac{C}{{\;C}_{0}}\right)\times 100\%$$
where R (%) is the removal rate of HM on the adsorbent; C_0_ (mg/L) is the initial concentration of HM; C (mg/L) is the remaining concentration of HM at each time increment.

The adsorption capacities (q_e_) of phosphates toward HM were calculated by Equation (2). The adsorption data was also simulated by the pseudo-first-order (Equation (3)) and pseudo-second-order (Equation (4)) models to analyze the adsorption kinetics of HMs [17].
(2)$$q_{e}=V\left(C_{0}-C_{e}\right)/m$$
(3)$$\ln(q_{e}-q_{t})=\ln q_{e}-k_{1}t$$
(4)$$\frac{t}{q_{t}}=\frac{t}{q_{e}}+\frac{1}{k_{2}q_{e}^{2}}$$
where q_e_ (mg/g) is the adsorption capacity of a HM onto the adsorbent; C_e_ (mg/L) is the residual concentration at which adsorption equilibrium is reached; q_t_ (mg/g) is the adsorption capacity of a HM onto the adsorbent at time t (h); k_1_ (h^−1^) is the pseudo-first-order rate constant; k_2_ (h^−1^) is the pseudo-second-order rate constant; V (mL) is the volume of solution used; and m (mg) is the mass of material used.

By comparing the adsorption capacities of copper, three superior phosphates were selected for further adsorption experiments with zinc. These zinc experiments were conducted similarly to the adsorption experiments with copper. Then, the adsorption characteristics of zinc were analyzed and compared to the results with copper. This work offered a foundation for the next experiments.

### 2.3. Selective Adsorption Experiments

To investigate the effect of mixed HMs on adsorption performance of each HM, the selective adsorption experiments were conducted in a mixed copper–zinc ions solution. The same masses of those three materials and treatment as the adsorption experiments (Section 2.2) were used to adsorb both Cu^2+^ and Zn^2+^ simultaneously. On the basis of the discrepant removal rates, selectivity of adsorbent toward each HM was evaluated by Equation (5).
(5)$$K_{s}=\frac{R_{2}}{R_{1}}-1$$
where K_S_ is selectivity coefficient for a specific metal ion, R_1_ is the removal rate of HM in single ion solution, and R_2_ is the removal rate of HM in mixed ions solution. The value of coefficient in comparison to 0 represents different selectivity: K_S_ > 0 means improved selectivity, K_S_ < 0 means inhibited selectivity, and K_S_ ≈ 0 means no change in selectivity. 

### 2.4. Desorption Experiments

The aim of desorption experiments was to investigate the stability of the adsorption onto the phosphates. At the end of the adsorption experiments of copper, the solid residues in flasks were vacuum-filtered and oven-dried at 80 °C to a constant weight for further desorption. Dried residues which absorbed Cu^2+^ were repeatedly added into 1000 mL DI water to desorb Cu^2+^. The stripped solution samples (10 mL) were pipetted into a 50 mL centrifuge tube after 0, 6, 12, 24, 48, and 72 h, to measure the concentration of Cu^2+^. Additionally, after continued desorption for 30 days, the influence of pH value on HM desorption was determined by detecting the content of Cu^2+^ in various pH conditions, ranging from 2.5 to 8.5, adjusted by the addition of dilute acid or alkali solutions.

## 3. Results and Discussion

### 3.1. Adsorption Performances of Copper on Different Materials

In the preliminary batch experiments, six trial materials were tested for the adsorption efficiency of copper. Results are shown in Figure 1. The copper removal occurred with different efficiencies depending on the phosphate materials. Little or no adsorption of copper was seen in CPM treatment, which indicates that it was seemingly unsuitable to capture copper metal. Encouragingly, the concentration of Cu^2+^ declined at different levels after adding HAP, BM, CP, CPH, and PAS, which showed positive adsorption effect on HMs. Particularly, the concentration of Cu^2+^ in PAS treatment decreased sharply during 0 to 6 h, and the maximum removal rate nearly reached 100%. Unlike other P-source materials, phytic acid sodium salt hydrate (PAS) showed remarkable performance, possibly contributed by its unique chemical molecule structure of six phosphate groups that are beneficial to chelate metals [25]. These fast removal outcomes suggested that phosphates are beneficial for large-scale treatment of metal ions.

### 3.2. The Removal Efficiency and Adsorption Kinetics of HMs on Three Phosphates

To further evaluate the adsorption performances, three phosphate adsorbents, including HAP, CP, and PAS, were selected to implement the next experiment of zinc removal. The removal performances of copper and zinc on three phosphates (HAP, CP, PAS) are shown in Figure 2. The maximum copper removal rate of HAP, CP, and PAS at the end of experimental time (72 h) reached 81.6%, 66.9%, and 98.8%, respectively (Figure 2a), and the maximum zinc removal rate of HAP, CP, and PAS reached 90.8%, 70.5%, and 2.0%, respectively (Figure 2b). The removal rates of copper were slightly lower than zinc in HAP and CP treatments, but both of them obtained great removal rates (>60%) of copper and zinc in the single metal ion solution. However, a distinct result occurred in PAS treatment, which had a better effect on copper removal, but very little effect on zinc removal. This result is very surprising, with such a large deviation between the good chelating effect of PAS with copper, but almost no chelating effect toward zinc. The data in Figure 2 show that the best adsorbent of copper was PAS, while the best adsorbent of zinc was HAP. Cao et al. [26] believe that the interaction between phosphate and zinc is mainly realized through surface adsorption and complexation. Cao used apatite ore powder to carry out solid–liquid interface reaction with heavy metal zinc, and their research results show that zinc fixed by surface adsorption or complexation reaches 95.7%, and there is no precipitation between zinc and phosphate. Through a series of experiments, Liu et al. [27] proved that iron phosphate nanoparticles can effectively reduce the solubility and biological activity of Cu^2+^ under neutral, acidic, and alkaline conditions. The solubility of Cu^2+^ decreases by 63~87%, while the biological activity of Cu^2+^ decreases by 54~69%. It is proved that phosphate can effectively reduce the activity of heavy metals in soil. As a result, appropriate adsorbents should be chosen carefully in practical applications, according to the main metal ions in the effluents.

HMs adsorption processes were demonstrated by the pseudo-first-order and pseudo-second-order kinetic models (Table 2). Since PAS had little adsorption effect on zinc, there is no model analysis of it. In terms of copper adsorption, the correlation coefficient (R2) of the pseudo-first-order kinetic model was lower than that of the pseudo-second-order model, which was contrary to that of zinc adsorption. Overall, higher coefficients were obtained in the first model (0.981–1.000) than the second model (0.979–1.000). Furthermore, fitted adsorption values (q_e_) obtained from the former were similar with the experimental results, while the fitted results from the latter were slightly higher than the experiment. These results suggest that the pseudo-first-order model was more befitting to describe the adsorption kinetics of the HMs onto phosphates. As shown in Figure 3, the pseudo-first-order kinetic models of HMs were established to further visualize the adsorption process.

### 3.3. Adsorption Selectivity of Phosphates in Mixed HM Ions Solution

Generally, multiple HMs are nearly always discovered in polluted environments, which leads to more complicated and difficult treatment for metal removal and remediation. Absorption of HMs is probably influenced by coexisting metal ions in a mixed environment, since they would competitively bind the adsorption sites with each other [17]. Therefore, the performance of phosphates needs to be evaluated with the adsorption selectivity in mixed-metal solution. Figure 4 shows the selective adsorption of two ions on three phosphates in mixed Cu^2+^–Zn^2+^ solution. These results indicate a preference for the selective adsorption of copper, because the maximum removal rates of copper in all groups are much higher than zinc removal. The HAP treatment showed 80.0% Cu^2+^ removal versus 45.2% removal for Zn^2+^ (Figure 4a); CP treatment showed 81.1% Cu^2+^ removal versus 30.0% Zn^2+^ removal (Figure 4b); PAS treatment showed 99.5% Cu^2+^ removal versus 6.7% Zn^2+^ removal (Figure 4c). Interestingly, maximum removal rates of zinc in mixed solution with HAP and CP treatment showed a dramatic decline, compared to the single zinc solution (Figure 2b), and the specific drops were 52.1% and 60.3%, respectively. This means that a great competitive adsorption occurred between copper and zinc in the mixed solution, and better adsorption affinity for the former. Previous research has shown that the affinity between heavy metal ions and adsorbents is closely related to the charge, radius, and electronegativity of heavy metal ions [28]. 

In addition, similar results (high removal rates of Cu^2+^ and few of Zn^2+^) in PAS treatment obtained from both single and mixed solution (see Figure 2 and Figure 4c) implies a little competition of each HM on PAS. In other words, the removal efficiency of copper on PAS was not obviously impacted by the presence of zinc. This competitive removal selectivity of HMs probably contributes to their competition on different combination sites.

The selectivity of HMs also was visualized by distribution diagram, as shown in Figure 4d. The preferred adsorption of Cu^2+^ was essentially unaffected by the selectivity of the three treatments, while Zn^2+^ adsorption experienced “reverse” selectivity in HAP and CP treatment. Although a considerably large, improved selectivity coefficient (Ks) of Zn^2+^ “appeared” to occur in PAS treatment, it is meaningless because of its low adsorption efficiency. Based upon these results, the conclusion can be drawn that adsorption selectivity is beneficial for using phosphates to remove copper in a mixed ions environment.

### 3.4. Adsorption Stability of HM on Phosphates

Stability is a very important parameter to evaluate the adsorption effect. Adsorption stability of copper was tested on three phosphate materials including HAP, CP, and PAS since all of these effectively decreased the Cu^2+^ concentration (Figure 1). Residues that adsorbed copper were dried and placed into DI water to desorb and measure the adsorption stability. As shown in Figure 5a, the dissociated content of Cu^2+^ increased slowly with time during the first 72 h and then dropped down over time in HAP and CP treatments but kept rising in PAS treatment with experimental time (until 720 h). The reason possibly is that PAS was destroyed and lost sectional absorption capability over time. 

Low pH environment is possible to influence the adsorption process due to the lessening of negatively charged sites by protonation [17]. In order to investigate the effect of pH on desorption equilibrium, the content of Cu^2+^ in desorption samples was detected with a wide range of pH (2.5 to 8.5). As shown in Figure 5b, a lower desorption was found in the initial environment (pH = 6.0) compared to both of the acidic and alkali treatments, which indicated that desorption of HM could be affected in different pH values. Despite these negative effects, the influence of time and pH on desorption was weak, because the total desorption quantity of copper was relatively low compared to the preceding adsorption quantity. Thus, these experimental P-source adsorbents showed synthetically great adsorption stability for HMs. This study can provide a basis for the application of HMs removal in manmade wetland systems, because they are able to remain stable in both acid and alkaline wastewater, even during long-term operation.

### 3.5. Potential Applications of Phosphates and Design of a Constructed Wetland System for HMs Removal

Although phosphate materials are widely used as effective amendments in literature, they have not been researched and applied yet in constructed wetlands. High removal efficiency and stability of phosphates toward HMs from affected water were obtained in this study. Therefore, these results indicate that this work is worthy of further study, to consider their application for HMs removal and immobilization from constructed wetlands. In view of the properties of phosphates, regarding chemical stability, thermal stability, and potential of HMs treatment [29], they show potential value for further development and application.

As well as in this study, some literature references have reported promising great sorption effects of phosphate adsorbents on HMs. These results are shown not only for single-metal aqueous solutions [30,31] but also multi-component metals in mine wastewater [32]. In addition, in the meantime, Saavedra-Mella et al. [33] found that phosphate treatment could alleviate the acute phytotoxicity of HMs. Hence, phosphate materials have potential application to be perceived as filters in wastewater pretreatment to reduce the loads of HMs in inflow, to minimize their toxic influence on the hydrophytes of the constructed wetlands.

Besides use in wastewater treatment, P-source materials are also studied, at present, for HMs immobilization in the fields of soil remediation [34] and sewage sludge composting [35,36,37]. According to many previous research studies, phosphate-based materials have been proven to have an strong capacity to immobilize HMs by preventing migration, improving the residue proportion, and reducing bioavailability [38,39]. Generally, this immobilization outcome arises from adsorption and precipitation reactions [40]. In addition, data regarding phosphates have been published, indicating that they were able to effectively reduce the availability and solubility of metal ions, via declining water solubility bioaccessibility and phytoavailability [22]. Sun et al. [41] concluded that immobilization of HMs can be attributed to the formation of amorphous phosphate rather than crystalline compounds, since heavy metals are chemical elements, and they cannot possibly be “biodegraded” as can organic pollutants. Thus, this phosphate treatment is an important strategy to achieve soil remediation. Moreover, in combination with other materials (such as biochar and nanoparticles), phosphate treatment may be a promising method to improve HMs remediation [27,42]. Wang et al. [43] summarized nearly all the substrates found in constructed wetlands, wherein apatite (Ca_10_(PO_4_)_6_(OH,F,Cl)_2_), as a class of natural minerals, is stable and has good adsorption capability for P, N, and heavy metals. As a result, phosphates can be used as the raw materials for adsorbents and filters and may have a positive influence on HMs removal and sediment soil amendment in constructed wetlands. 

In general, constructed wetlands always contain multiple parts. As shown in Figure 6, a constructed wetland adsorption system comprises phosphate adsorbents, gravel, wastewater, soil, hydrophytes, and other components. In this system, contaminants from inflow will be adsorbed, first, in the pretreatment process. Then, nutrients will be removed by the plants, and residual ions will be immobilized into phosphate minerals in the soil sediment. According to the data of the adsorption experiments, equilibrium time should be about 72 h (Section 3.2). The flow rates at the inlet and outlet need to be controlled properly by valves to ensure effective HMs removal and water purification. Based on outstanding adsorption stability (obtained in Section 3.4), this system will allow treatment of multiple types of wastewater and run effectively for a long cycle (at least one month). After the completion of processing steps: chemical adsorption, biosorption, immobilization, and re-adsorption, effluents will have effectively achieved purification. At the end of these long-term operating cycles, metals can be recycled as oxide crystals by calcination and precipitation [44]. It is worthy to emphasize the advantages of this adsorption system for efficiency, security, and economy. In particular, using inexpensive and abundant phosphate mineral materials results in lower operating costs, which are more feasible for developing countries [45]. In addition, the process of HMs migration from the environment, and away from living organisms, will be effectively controlled, thus improving the security of manmade wetland ecosystems.

In summary, utilization of phosphate compounds for HMs removal is promising in constructed wetlands, because of ease of operation, remarkable HM removal effects, low operating costs, and the harmlessness of the resulting compounds. There are three aspects of this application: (1) phosphate adsorbents effectively remove HMs from wastewater; (2) they reduce the toxic risk for plants by decreasing the loads of contaminants; (3) they are also able to immobilize and recycle HMs to accomplish wetland soil remediation. With the help of phosphates, this constructed wetland adsorption system, as shown in (Figure 6) will become more effective and multifunctional for HMs removal.

## 4. Conclusions

In this study, several phosphate compounds were tested for the feasibility of HMs removal from aqueous solutions. PAS showed prominent removal performance for copper, but little for zinc. HAP and CP showed the highest removal efficiencies for both copper and zinc in a single metal ion solution. In particular, phosphates have a higher selectivity to remove copper rather than zinc, in a solution that contains both Cu^2+^ and Zn^2+^. The adsorption kinetics of HMs were effectively simulated by pseudo-first-order models. Moreover, the adsorption effect of phosphates for HMs was stable, and exhibited only weak desorption, even after 30 days, under a wide range of pH environment. Removal efficiency of HMs obviously varied, depending on the types of both HMs and phosphates present, with ranking orders as follows: copper > zinc, PAS > HAP > CP. On the basis of the potential for phosphates to remove HMs, this paper also provides a hypothetical application for metal removal and immobilization in the field of constructed wetlands. According to the structural advantages of phosphate itself, it can effectively reduce or delay the clogging situation; on the other hand, it shows a better adsorption effect on heavy metals and stability, which is expected to become a new choice of artificial wetland filler. Despite the encouraging results presented in this paper, the practical influence of phosphates on constructed wetlands is still unknown. Indeed, it will be necessary to perform additional research with actual constructed wetlands application testing to promote the use of this promising remediation technology in the future.

## Figures and Tables

**Figure 1 ijerph-19-05344-f001:**
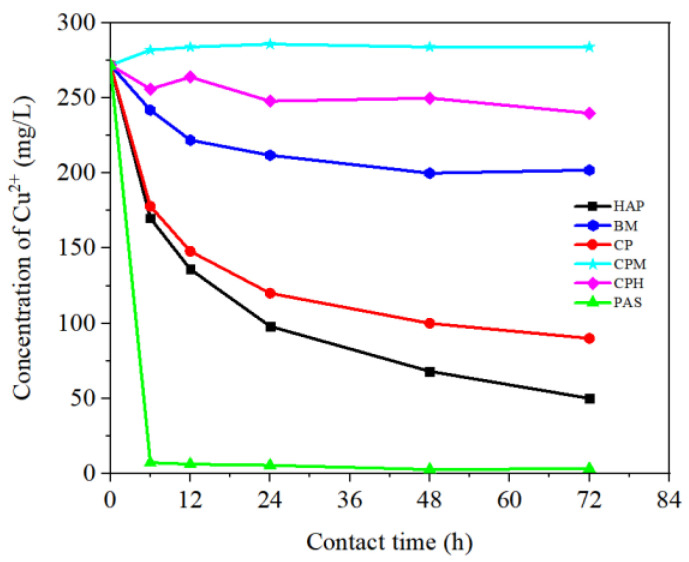
The concentration variation of Cu^2+^ in six phosphate treatments within 0 to 72 h.

**Figure 2 ijerph-19-05344-f002:**
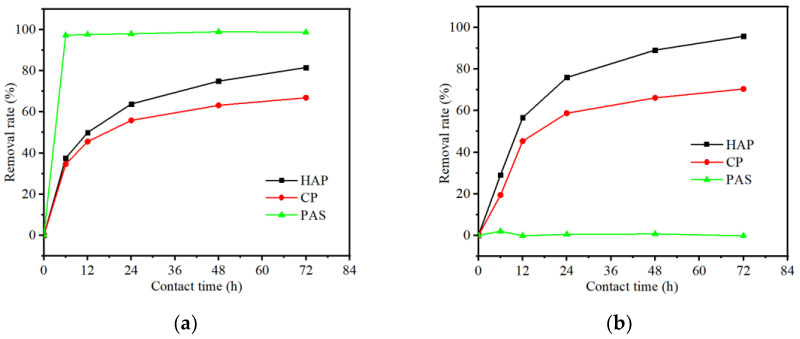
The removal performances of HMs on three phosphates (HAP, CP, PAS) in aqueous solution, (**a**) Cu^2+^ removal, and (**b**) Zn^2+^ removal.

**Figure 3 ijerph-19-05344-f003:**
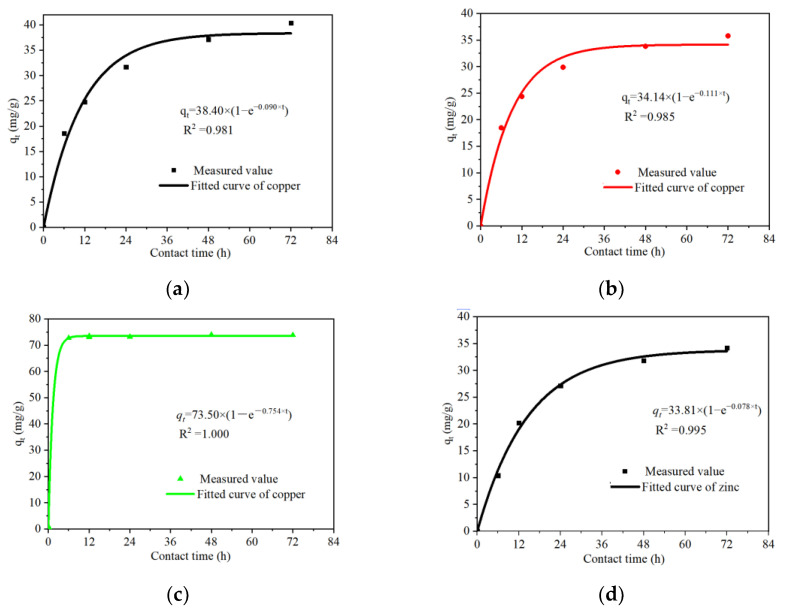
The pseudo-first-order kinetic models of HMs adsorption, Cu^2+^: (**a**) HAP, (**b**) CP, (**c**) PAS; Zn^2+^: (**d**) HAP, (**e**) CP.

**Figure 4 ijerph-19-05344-f004:**
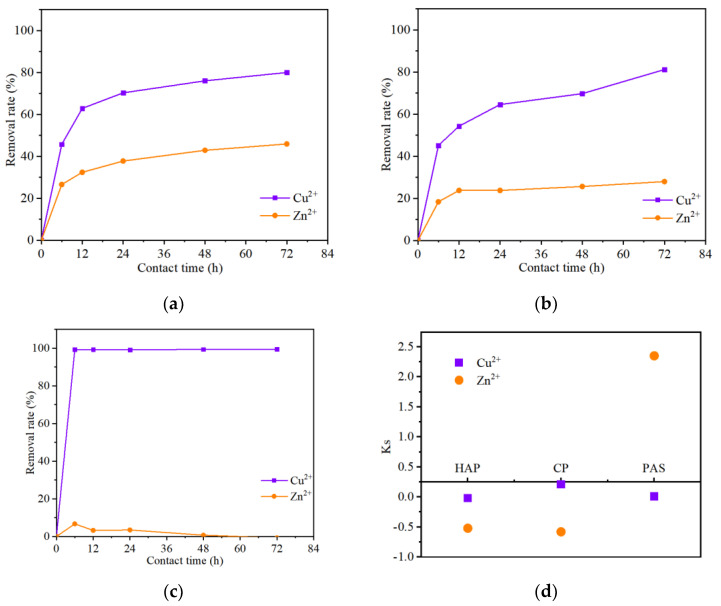
The selective adsorption of Cu^2+^ and Zn^2+^ on HAP (**a**), CP (**b**), PAS (**c**); the distribution of selectivity coefficient (**d**).

**Figure 5 ijerph-19-05344-f005:**
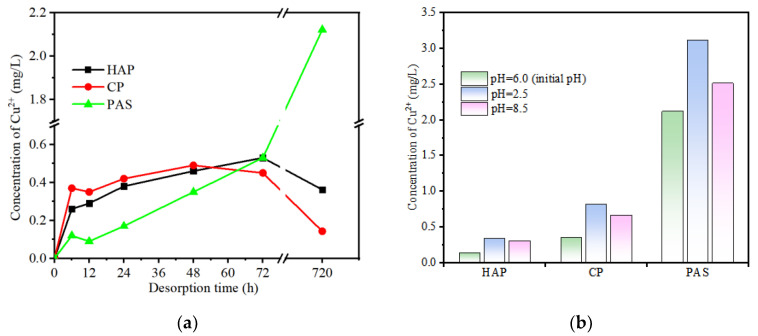
Influence of desorption time (**a**) and pH (**b**) on stability of adsorption.

**Figure 6 ijerph-19-05344-f006:**
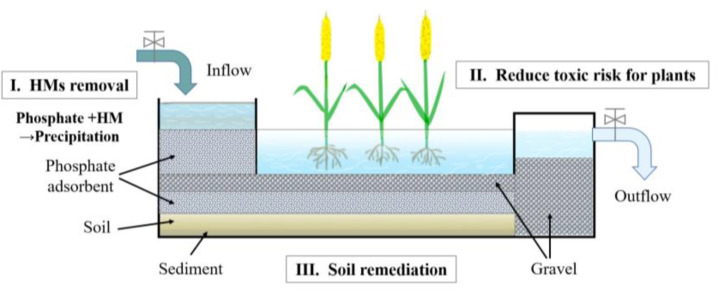
Schematic diagram of phosphates’ potential applications in a constructed wetland system.

**Table 1 ijerph-19-05344-t001:** Chemical characteristics of experimental phosphate materials.

Phosphate Source	Abbreviation	Chemical Composition	Molecular Weight	Purity	Purity Level	Phosphorus Content
Hydroxyapatite	HAP	Ca_5_(PO_4_)_3_OH	502.31	≥96%	AR	17.77%
Bone Meal	BM	Ca_3_(PO_4_)_2_	310.18	≥20%	-	4.00%
Calcium Phosphate	CP	Ca_3_(PO_4_)_2_	310.18	≥96%	AR	19.18%
Calcium Phosphate monobasic Monohydrate	CPM	Ca(H_2_PO_4_)_2_·H_2_O	252.06	≥85%	AR	20.91%
Calcium Phosphate Hydrate	CPH	Ca(H_2_PO_4_)_2_·_X_H_2_O	234.05 (as anhydrous)	≥92%	AR	24.37%
Phytic Acid Sodium	PAS	C_6_H_18_O_24_P_6_·xNa + yH_2_O	660.04 (anhydrous free acid basis)	≥99%	AR	27.89%

Note: “-” none of data.

**Table 2 ijerph-19-05344-t002:** Kinetic parameters of pseudo-first-order and pseudo-second-order models for adsorption of copper and zinc by phosphates.

HMs	Materials	Pseudo-First-Order Model			Pseudo-Second-Order Model		
		q_e_ (mg/g)	k_1_ (h^−1^)	R^2^	q_e_ (mg/g)	k_2_ (g/mg/h^−1^)	R^2^
Cu (II)	HAP	38.40	0.090	0.981	45.45	0.0023	0.998
	CP	34.14	0.111	0.985	39.22	0.0035	1.000
	PAS	73.50	0.754	1.000	74.07	0.0870	1.000
Zn (II)	HAP	33.81	0.078	0.995	41.49	0.0016	0.994
	CP	27.00	0.074	0.983	33.11	0.0000	0.979
	PAS	-	-	-	-	-	-

Note: “-” none of data.

## Data Availability

The data presented in this study are available on request from the corresponding author.

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
