# Peer review of "Application of Phosphate Materials as Constructed Wetland Fillers for Efficient Removal of Heavy Metals from Wastewater"

_ijerph, 2022, doi:10.3390/ijerph19095344_

Round 1
Reviewer 1 Report
This research deals with the constructed wetlands as one of the sewage treatment technology for the removal of heavy metals Cu and Zn. Several phosphate materials were tested to remove these heavy metals.
Although, the paper is very interesting there are some issues that need to be solved.
The contribution and explanation should be presented better. Why authors decided to use the proposed materials and methodology? Is there any other research on the used phosphate materials so that the comparison can be explained? More review of the literature should be given in the Introduction and Discussion should be provided.
What are the future directions of the research? What about other heavy materials, how can the proposed methodology be applied to their removal?
Reviewer 2 Report
The interst on constructed wetlands as environmentally friendly and economically efficient sewage treatment is gowing in the last years again. It is well known they can be used for the removal of heavy metals which is a topic of higher concern each day due to their high presence in the ecosystems and groundwaters. It is clear that the fillers of constructed wetlands play an important role in heavy metals retention. Therefore the topic is of interest.
However the manuscript lacks the novelty required for its publication. It is more a report on the adsoption of two metals by different fillers following the steps given in many other publications. It lacks the scientific sound or progress beyond the state of the art for its publication.
Therefore I cannot recomend to publish it in IJERPH
Reviewer 3 Report
The manuscript “Application of phosphate materials as constructed wetland fillers for efficient removal of heavy metals from wastewater” deals with an interesting application of phosphate materials for heavy metals removal from wastewater. The research is novel, and a deep and meticulous study was carried out by the authors. However, for the paper, minor revisions are required.
- Materials and methods
- (Pag. 3 Lines 106-107) “Then, the concentration of Cu in 106 each sample were tested according to the method described by”. Add the method author.
- (Pag. 3 Lines 109-110) Error for the equation reference.
- (Page 4 Line 2) All the variables need to be described through the text. For example, what is “Ce” into the equation 2?
- (Page 4 Line 123) What is “w”?
- (Page 4 Line 126) What is “E”?
- (Page 4 Line 142) Subsection 2.2.1 does not exist through the manuscript.
- Results and discussion
- (Pag. 5 Line 166) Error for the figure reference.
- (Pag. 5 Line 177) Fig.1 shows the observed results for the copper concentration. Why the concentration was not reported for the zinc too? In Figure 2, for both copper and zinc removal efficiencies were reported. only the change in the concentration of copper reported and not that of zinc?
- (Pag. 5 Line 183) Error for the figure reference.
- (Pag. 6 Line 196) Change figure caption “Figure 1” to “Figure 2”.
- (Pag. 6 Line 207) Error for the figure reference.
- (Pag. 7 Line 216) Change figure caption “Figure 2” to “Figure 3”. From this point on, all the figures capture need to be corrected.
- (Page 9 Lines 280-285) May phosphates be used for other application, such as for example the removal of nutrients from agricultural practices? For instance "Diffuse Water Pollution from Agriculture: A Review of Nature-Based Solutions for Nitrogen Removal and Recovery. Mancuso, G., Bencresciuto, G. F., Lavrnić, S., & Toscano, A. | Water, 13(14), 1893 (2021)”. Please, discuss it through the text.
Reviewer 4 Report
Review
The manuscript entitled „Application of phosphate materials as constructed wetland fillers for efficient removal of heavy metals from wastewater” presents the study about heavy metals (HMs) removal from wastewater. For this purpose, several phosphate materials were tested to remove the Cu and Zn from aqueous solution. As a result of the study conducted by the Authors, the most remarkable efficiency for heavy metal removal among tested phosphates was PAS. Further research may provide a basis for the application of HMs removal in the man-made wetland systems.
Concerning the abstract (and beyond) – I personally do not believe that any substance or compound described is „excellent”. "Excellent" occurs 7 times in the text – this is not scientific language. In my opinion, no scientific research (in any field) confirms the excellence of anything. A big plus for the GA.
The methodology is concise and the results are promising, but the article lacks conclusions supported by the research conducted.
Summarising, the manuscript is written not quite correct – Chapter 4 and partly Chapter 3 do not follow the form of a scientific paper. My recommendations is to accept this work after some revisions (see „Major remarks” and „Minor mistakes and shortcomings” below).
Best regards,
Reviewer
Major remarks:
1.) Results and Discussion: the information presented in this chapter is chaotically gathered in several subchapters, some of which are so short as to be unnecessary - they do not add any value (e.g. 3.5. – 3.5.1., 3.5.2. and 3.5.3.);
2.) Conclusions:
- a) the conclusions are written in too general terms - perhaps the most important achievements should be highlighted?, and, in fact, if you think about it, there are no clear conclusions in the Conclusions section;
- b) against the background of the whole work, the last chapter seems too short - underdeveloped and hastily written.
Minor mistakes and shortcomings:
- the Keywords also appear in the title of the article; it is better to avoid such repetition and change some of the repeated words while leaving the title unchanged;
- lines 109-110 - an error has crept in; (line 123 and 126 „w”, „E”?... line 166, 183, 207-208, 224, 228-229, 230, 231, 233-234, 240-241, 258-259);
- p. 5 - Figure 1, p. 6. also Figure 1 (second Figure No. 2 = Figure 3 and 3=4?).
Round 2
Reviewer 1 Report
The authors have addressed all comments.